# A Pilot Study on Plasma and Urine Neurotransmitter Levels in Children with Tic Disorders

**DOI:** 10.3390/brainsci12070880

**Published:** 2022-07-04

**Authors:** Qiao-Qiao Qian, Qian-Qian Tan, Dan Sun, Qing Lu, Ying-Ying Xin, Qian Wu, Yong Zhou, Yang-Xi Liu, Pei-Chao Tian, Zhi-Sheng Liu

**Affiliations:** 1Department of Neurology, Wuhan Children’s Hospital of Tongji Medical College, Huazhong University of Science and Technology, Wuhan 430000, China; qianqiaoqiao1983@163.com (Q.-Q.Q.); bloveriver@163.com (D.S.); lqingqing14@163.com (Q.L.); xyy2038330319@163.com (Y.-Y.X.); wuqian20140617@163.com (Q.W.); 2Department of Marketing, Wuhan Kindstar Clinical Diagnostic Institute Co., Ltd., Wuhan 430000, China; tqqkl2007@126.com (Q.-Q.T.); zhouyong8@kindstar.com.cn (Y.Z.); liuyangxi@kindstar.com.cn (Y.-X.L.); 3Department of Pediatrics, First Affiliated Hospital of Zhengzhou University, Zhengzhou 450052, China

**Keywords:** tic disorders, Tourette syndrome, neurotransmitter

## Abstract

Background: Tic disorders (TDs), including Tourette syndrome, are childhood-onset neuropsychiatric disorders characterized by motor and/or vocal tics that commonly affect children’s physical and mental health. The pathogenesis of TDs may be related to abnormal neurotransmitters in the cortico-striatal-thalamo-cortical circuitry, especially dopaminergic, glutamatergic, and serotonergic neurotransmitters. The purpose of this study was to preliminarily investigate the differences in the three types of neurotransmitters in plasma and urine between children with TD and healthy children. Methods: We collected 94 samples of plasma and 69 samples of urine from 3–12-year-old Chinese Han children with TD before treatment. The plasma and urine of the same number of healthy Chinese Han children, matched for age and sex, participating in a physical examination, were collected. Ultra-performance liquid chromatography-tandem mass spectrometry was used to detect the three types of neurotransmitters in the above samples. Results: The plasma levels of norepinephrine, glutamic acid, and γ-aminobutyric acid, and the urine levels of normetanephrine and 5-hydroxyindoleacetic acid were higher in the TD children than in healthy children. The area under the curve (AUC) values of the above neurotransmitters in plasma and urine analyzed by receiver operating characteristic curve analysis were all higher than 0.6, with significant differences. Among them, the combined AUC of dopamine, norepinephrine, normetanephrine, glutamic acid, and γ-aminobutyric acid in the 8–12-year-old subgroup was 0.930, and the sensitivity and specificity for TD were 0.821 and 0.974, respectively (*p* = 0.000). Conclusions: There are differences in plasma and urine neurotransmitters between TD children and healthy children, which lays a foundation for further research on the pathogenesis of TD.

## 1. Introduction

Tic disorders (TDs), including Tourette syndrome (TS), are childhood-onset neuropsychiatric disorders that are mainly characterized by sudden, rapid, repeated, and rhythmless motor tics and/or vocal tics [1]. TD is commonly divided into transient TD (TTD), chronic motor or vocal TD (CTD), and TS. The prevalence of TTD (1.2%), CTD (0.9%), and TS (0.4%) in China is lower than the reported worldwide prevalence (TTD 2.99%, CTD 1.61%, and TS 0.77%). The overall prevalence of TD in China is 2.5% [2,3,4,5]. TD is often comorbid with attention-deficit/hyperactivity disorder, obsessive-compulsive disorder, and anxiety/depression disorders. The pathogenesis of TD is complex and unclear, but neuroanatomical and neurophysiological studies have found neurotransmitter abnormalities in the cortico-striatal-thalamo-cortical circuitry (CSTC). Neurotransmitter detection in the cerebrospinal fluid (CSF), plasma, and urine of animal models, responses to drug treatments that block or stimulate neurotransmitters, autopsy results, and neuroimaging studies have found that neurotransmitter abnormalities mainly involve dopaminergic, glutamatergic, and serotonergic neurotransmitters [1,6]. Moreover, previous studies of the profile of neurotransmitters in blood of TS or TD individuals were mainly conducted in serum, involving dopamine (DA), norepinephrine (NE), glutamic acid (Glu), γ-aminobutyric acid (γ-GABA), etc. [7,8,9,10]. In this study, differences in these neurotransmitters in plasma and urine between TD children and healthy children were investigated. The results of this study can provide guidance for further exploring the pathogenesis of TD.

## 2. Materials and Methods

### 2.1. Chemicals and Reagents

Methanol and acetonitrile were HPLC grade and purchased from Sigma–Aldrich (Shanghai, China). Methanic acid (88% and 98%), concentrated hydrochloric acid (12 mol/L), and ammonium acetate were purchased from Sinopharm Chemical Reagents Co., Ltd. (Shanghai, China). The experimental water was purchased from Yichang Wahaha Qili Food Co., Ltd. (Yichang, China).

DA, epinephrine (E), and 3-methoxytyramine were purchased from Sigma–Aldrich. NE, 5-hydroxytryptamine (5-HT), γ-GABA, Glu, tyrosine (Tyr), tryptophan (Trp), vanillylmandelic acid (VMA), homovanillic acid (HVA), 5-hydroxyindoleacetic acid (5-HIAA), and creatinine were purchased from Shanghai YuanLeaf Biotechnology Co., Ltd. (Shanghai, China). Metanephrine (MN) was purchased from Aladdin (Shanghai, China). Normetanephrine (NMN) was purchased from Toronto Research Chemicals (Toronto, ON, Canada).

The isotope-labelled internal standards (IS), DA-D4, NE-D6, and NMN-D3, were purchased from Cambridge Isotope Laboratories, Inc. (Boston, MA, USA). Trp-D5 and 5-HIAA-D2 were purchased from Shanghai Zhenzhun Biotechnology Co., Ltd. (Shanghai, China). 5-HT-D4 and creatinine-D3 were purchased from Sigma–Aldrich.

### 2.2. Patients and Control Groups

In this prospective study, the patients attended the outpatient or inpatient department of Wuhan Children’s Hospital and the First Affiliated Hospital of Zhengzhou University between 1 January 2021 and 31 December 2021. The inclusion criteria for participants were as follows: met the diagnostic requirements for TD in the American Diagnostic and Statistical Manual of Mental Diseases, fifth Edition (DSM-5), were between 3 and 12 years of age, and were of Chinese Han nationality. The exclusion criteria included those with other psychiatric disorders or mental motor retardation, those taking psychiatric medications, those with a history of infectious disease in the previous 2 weeks, and other unfit participants. In parallel, a control group was recruited at Wuhan Children’s Hospital and the First Affiliated Hospital of Zhengzhou University. The control group consisted of children aged 3–12 years who were of Chinese Han nationality and prepared for a physical check-up. After the examination, those with neurological diseases, infectious diseases, or a history of drug administration in the previous 2 weeks were excluded. These healthy children were included as participants in control groups. The control groups were fully matched to the age and sex of the patient groups. All subjects recruited for this study gave written informed consent prior to entering the study. The study was approved by the Ethics Committee of Wuhan Children’s Hospital.

Previous unpublished results from our team showed that the levels of neurotransmitters from plasma and urine were not significantly different between men and women in a healthy population, whereas some neurotransmitters significantly differed between ages. The enrolled subjects were divided into 3–7-year-old and 8–12-year-old groups. The numbers of each group are shown in Table 1. The detected sample size and detection rates of each neurotransmitter from plasma and urine are shown in Appendix A. Further analysis of MN and 3-methoxytyramine in plasma was not performed because the detection rates of both neurotransmitters were less than 15%.

### 2.3. Samples

Plasma or urine was collected from the enrolled patients with TD and control children. The participants needed to provide a sample on an empty stomach between 8:00 and 10:00 am. They were seated for more than half an hour before sampling to avoid the excitement caused by exercise. Blood (2–3 mL) was collected in an EDTA tube, and a 2–3 mL random sample of urine was collected in a special sterile tube. The samples were collected and transported at a cold temperature and immediately sent for inspection. All samples were measured at the mass spectrometry laboratory of Wuhan Kindstar Diagnostics Co., Ltd., Wuhan, China.

### 2.4. Sample Preparation

All IS, standard product, quality control, patient samples, and reagents were equilibrated to room temperature for at least 15 to 30 min. Plasma sample DA, E, NE, MN, NMN, and 3-methoxytyramine were extracted using solid phase extraction, and plasma sample 5-HT and γ-GABA, Glu, Tyr, and Trp and urine neurotransmitters were extracted using protein precipitation. The sample procedures were optimized based on the recently reported research results [11,12,13].

### 2.5. UPLC–MS/MS and Quantification

The analyses were conducted on a Shimadzu ultra-performance liquid chromatograph with a Shimadzu LCMS-8050 triple quadrupole mass spectrometer (Shimadzu, Kyoto, Japan). MS/MS detection was performed in positive or negative electrospray ionization mode using multiple reaction modes (MRM). Forty microliters of the clean supernatant or filtered solution was placed into an autosampler vial, and 10 μL of liquid was injected into the system. Using the LabSolutions data processing system, version 5.81 (Shimadzu, Kyoto, Japan), the standard concentration was the horizontal coordinate, the standard and internal peak area ratio was the vertical coordinate, linear regression was weighted (w = 1/C^2^) or (W = 1/C), and the result of R^2^ was >0.99.

### 2.6. Statistical Analysis 

All experimental data were analyzed by SPSS version 19.0 (SPSS Inc., Chicago, IL, USA). Shapiro–Wilk analysis was used to assess the normality of the distribution. Due to the nonnormal distribution, the 2.5th, 50th, and 97.5th percentiles were calculated for each group. A nonparametric independent sample t-test was used for difference analysis. The diagnostic performance of neurotransmitters for TD was calculated by receiver operating characteristic (ROC) curve analysis. Univariate analysis was performed for all neurotransmitters, and then neurotransmitters with significant differences (*p* < 0.05) and areas under the curve (AUCs) greater than 0.5 were included in the multivariate analysis. The neurotransmitters included in the multivariate analysis were analyzed by regression analysis, and the results of the regression analyses were analyzed with a combined ROC curve analysis. In all analyses, *p* < 0.05 indicated a significant difference.

## 3. Results

### 3.1. Levels of Neurotransmitters from Plasma and Urine

The levels of neurotransmitters from plasma and urine in patients with TD and controls are shown in Table 2 and Table 3. Data from each of the groups were not normally distributed, so the level of neurotransmitters is shown as the median and 2.5th and 97.5th percentiles.

Plasma levels of NE, Glu, and γ-GABA were significantly higher in the TD group than in the control group of 3–7-year-old children. Plasma levels of DA, NE, NMN, Glu, and γ-GABA were significantly higher in the TD group than in the control group of 8–12-year-old children. Conversely, plasma levels of Trp were significantly lower in the TD group than in the control group of 8–12-year-old children. Urinary levels of NMN and 5-HIAA were significantly higher in the TD group than in the control group of 3–7-year-old children. Urinary levels of NE, MN, NMN, and 5-HIAA were significantly higher, whereas urinary levels of E and HVA were significantly lower, in the TD group than in the control group of 8–12-year-old children. With increasing age, the number of neurotransmitters showing significant differences in both plasma and urine increased.

### 3.2. Assessing the Predictive Value of Neurotransmitters for Predicting TD

All neurotransmitters from plasma and urine in the different age groups were tested by ROC curve analysis to assess the predictive value for predicting TD (Figure 1; Appendix A).

The ROC curve analyses indicated that NE, Glu, and γ-GABA levels in plasma were suitable for distinguishing between the TD children and controls among the 3–7-year-old children, with AUCs of 0.642, 0.661, and 0.715, respectively. The combined diagnostic ability of these three neurotransmitters was improved with an AUC that was increased to 0.723, with a sensitivity of 0.960 and a specificity of 0.600 (*p* = 0.007). The AUCs for DA, NE, NMN, Glu, and γ-GABA levels in plasma for the 8–12-year-old children were 0.688, 0.654, 0.720, 0.732, and 0.788, respectively. The AUC for the combination of these five neurotransmitters was increased to 0.930, with a sensitivity of 0.821 and a specificity of 0.974 (*p* = 0.000).

The detection of neurotransmitters in urine can also be useful in differentiating patients with TD from controls. The AUCs for NMN and 5-HIAA levels in urine from 3–7-year-old children were 0.696 and 0.718, respectively. The AUC for the combination of these two neurotransmitters was 0.696, with a sensitivity of 0.667 and a specificity of 0.750 (*p* = 0.002). The AUCs for NE, MN, NMN, and 5-HIAA levels in urine from 8–12-year-old children were 0.623, 0.714, 0.741, and 0.669, respectively. The AUC for the combination of these four neurotransmitters was 0.759, with a sensitivity of 0.644 and a specificity of 0.756 (*p* = 0.002).

The combined detection of neurotransmitters improved predictive values for differentiating patients with TDs and controls.

## 4. Discussion

The pathogenesis of TD is not completely clear to date. It is generally considered to be the result of an interaction between genetic and environmental factors. Available neurochemical, neuroimaging, animal experiments, drug treatment responses, and autopsy results all support a role for neurotransmitter abnormalities in the CSTC, and neurochemical analyses have mainly suggested dopaminergic, glutamatergic, and serotonergic neurotransmitters [1]. Detection of these neurotransmitters can theoretically be obtained from CSF, plasma, or urine. However, CSF tests require a lumbar puncture, a procedure that carries a risk of injury. For ethical reasons, it is difficult to collect CSF for testing from patients with TD or healthy individuals. Although the relationship between blood and brain is not direct, neurotransmitters can move or penetrate the blood–brain barrier with the assistance of neurotransmitter transporters and are driven by electrochemical gradients [14]. Therefore, plasma neurotransmitter levels may indirectly reflect changes in neurotransmitter levels in the central nervous system (CNS). Under the catalysis of enzymes, dopaminergic and serotonergic neurotransmitters in plasma can be converted into HVA and 5-HIAA, respectively, which are excreted from urine. Therefore, it is necessary to simultaneously assess levels of neurotransmitters in plasma and urine. In this study, the UPLC–MS/MS method was used to detect neurotransmitters in plasma and urine, and some differences in neurotransmitters were found between TD children and healthy children.

Evidence supporting DA abnormalities in TD from previous studies is extensive. Norsalsolinol (NSAL) regulates the transmission and metabolism of DA in the CNS in vivo. A case–control study reported by Capetian P in 2021 showed that the urinary concentrations of NSAL were significantly increased in patients with TS, indicating that dopaminergic hyperactivity is involved in the pathophysiology of TS [15]. In a study assessing changes in DA in TS patients by positron emission tomography (PET), Steeves and colleagues showed that, after deep brain stimulation, the clinical improvement was related to decreased DA release and upregulation of DA receptors [16]. Other studies showed that, after taking amphetamines, DA release in the putamen in TS patients was greater than that in controls [17]. Accordingly, Zhao M. et al. found that electrical high-frequency stimulation of the globus pallidus internus could modulate decreases in dopaminergic activity in the striatum in a rat model of TS [18]. All these studies suggest that DA may be involved in the pathogenesis of TD, and indeed, the clinical use of DA receptor blockers in the treatment of TD is effective, which also supports the role of DA in the pathogenesis of TD [19]. However, Abi-Jaoude E. et al. assessed DA levels in 11 TS patients and 11 matched healthy controls by PET and found no significant difference between the two groups [20]. This result challenges the hypothesis that striatal DA plays a major role in the pathophysiology of TD. Previous studies focused on DA concentrations in serum of patients with TD were inconsistent. Hu et al. found that the TD patients had significantly higher DA levels compared to controls [7], but opposite findings were reported by Xiao et al. [8]. In our study, no significant differences in DA were found between TD and controls when comparing groups of 3–7-year-old children, but significantly increased DA was found in the TD group in 8–12-year-old children. This may be because DA is not the only factor in the onset of TD, because age is one of the factors affecting abnormal DA secretion in TD children, and due to the differences between catatonic (low base) and staged (sudden discharge) DA release systems.

Moreover, noradrenergic fibers project broadly from the locus coeruleus to the cerebral cortex and modulate the subcortical circuitry involved in TS. It has been suggested that stimulation of central α2-adrenergic receptors can reduce the release of catecholamines [21]. Adrenergic agonists (e.g., clonidine and guanfacine) have been shown to be more effective than placebo in reducing the severity of tics [22]. Previous studies documented that serum NE levels are higher in children with TD than in healthy children [8,10]. The present study also found that plasma NE levels were significantly higher in children with TD than in healthy children in the different age groups. NE is metabolized to NMN in the body, and accordingly, we also found that urinary NMN levels were significantly higher in TD children than in healthy children. Overall, our findings also support a role for NE in the pathogenesis of TS.

Previous studies have mostly studied the levels of Glu and γ-GABA from imaging, and the conclusions have not been completely consistent. Regarding other neurotrasmitters assessed in relation with TD pathogenesis, Glu is another excitatory neurotransmitter. Naaijen J used the 3.0T NMR spectrum to study the difference in Glu in the dorsal striatum and anterior cingulate gyrus of children with TD, and no statistically significant differences were found, an issue which may be related to the inability of 3.0T NMR to distinguish Glu from glutamine [23]. However, another study in adults with TD showed a decrease in Glu and glutamine in the striatum and thalamus [24]. In addition, Harris and colleagues, using 7.0T MRI, revealed elevated Glu concentrations in the premotor cortex in TD patients but no significant difference in the striatum [25]. γ-GABA is an inhibitory neurotransmitter, and its role in TD has also been demonstrated. Through PET imaging, Lerner A. et al. found that the binding of γ-GABA in the bilateral ventral striatum, globus pallidus, and thalamus was decreased in TS [26]. McCairn K.W. et al. showed that the local destruction of GABA-ergic neurons in the striatum of monkeys produced a vocal tic [27]. Baclofen, a GABA receptor agonist, is also effective in the treatment of TD. A small number of reports that studied the levels of Glu and γ-GABA from serum showed that patients with TD or TS had higher Glu levels and lower γ-GABA levels compared to healthy children [9,10]. Our study found that plasma levels of Glu and γ-GABA were significantly increased in children with TD in different age groups. The increase in Glu and γ-GABA may be related to the interconversion of the two neurotransmitters in vivo.

Finally, serotonergic circuits in the brain are involved in reward processing, and 5-HT may also play an important role in the onset of TD. Niens J. et al. found that 5-HT mediates part of the role of DA by antagonizing DA, and DA denervation resulted in an increase in 5-HT in the basal ganglia [28]. 5-HT needs to be cleared in the body by 5-HT transporters, which are encoded by the SLC6A4 gene. Hildonen M. et al. studied the SLC6A4 gene in 57 TS patients and found that the mRNA expression of the SLC6A4 gene was significantly increased compared with that in the healthy control group [29]. In this regard, there were no significant differences in the plasma levels of 5-HT between TD children and healthy children in this study. The metabolite of 5-HT is 5-HIAA, which is excreted in urine, and we found that there were significant differences in 5-HIAA in the different age groups. The nonsignificant difference in plasma 5-HT concentrations may be related to increased 5-HT transporter activity, which may lead to the increase in 5-HIAA in urine, supporting the involvement of serotonergic systems in the onset of TD.

Previous studies of neurotransmitter concentrations in serum analyzed the differences in levels between TD and healthy control groups; the present study further analyzed the specificity and sensitivity of neurotransmitter detection in plasma and urine for the diagnosis of TD. ROC curve analysis can be used to select the best diagnostic threshold value for certain diagnostic methods. The larger the AUC value of ROC curve analysis, the higher its diagnostic value. In this study, ROC curve analysis of neurotransmitters in plasma and urine showed that the AUC with NE, Glu, and γ-GABA levels in plasma was between 0.642 and 0.788, and the AUC with NMN and 5-HIAA levels in urine was between 0.669 and 0.741. The combined analyses with multiple abnormal neurotransmitters improved the sensitivity and specificity in differentiating TD patients from controls. The results of this study further confirm that the pathogenesis of TD may be the result of the interaction of multiple neurotransmitters in vivo.

The present study has the following limitations: (1) The types of neurotransmitters detected were limited. There may be other neurotransmitter abnormalities in vivo that have not been detected; Müller-Vahl K.R. et al. found that cannabinergic neurotransmitters may play a role in the onset of TD and contribute to dopaminergic neurotransmitter abnormalities [30]. (2) The sample size was small. UPLC–MS/MS has been used for the detection of neurotransmitter concentrations in humans for a short time, and the number of patients recruited was limited (3) The children with TD were not diagnostically classified, and the Yale Global Tic Severity Scale score was not obtained. This did not allow us to determine whether there were differences in neurotransmitters among patients with different onset times, tic types, and tic severities. (4) Dynamic changes from before to after treatment were not assessed. Future studies overcoming all these limitations could further assess the diagnostic and prognostic value of circulating levels of differences in neurotransmitters in TD children.

## 5. Conclusions

In this study, the UPLC–MS/MS technique revealed differences in dopaminergic, glutaminergic, and serotonergic neurotransmitters in the plasma and urine of TD children vs. healthy controls. The results of ROC curve analyses showed that the combined diagnosis based on multiple neurotransmitters could be more valuable. The results of this study establish a rationale for further evaluating neurotransmitter circulating levels as a biomarker of TD.

## Figures and Tables

**Figure 1 brainsci-12-00880-f001:**
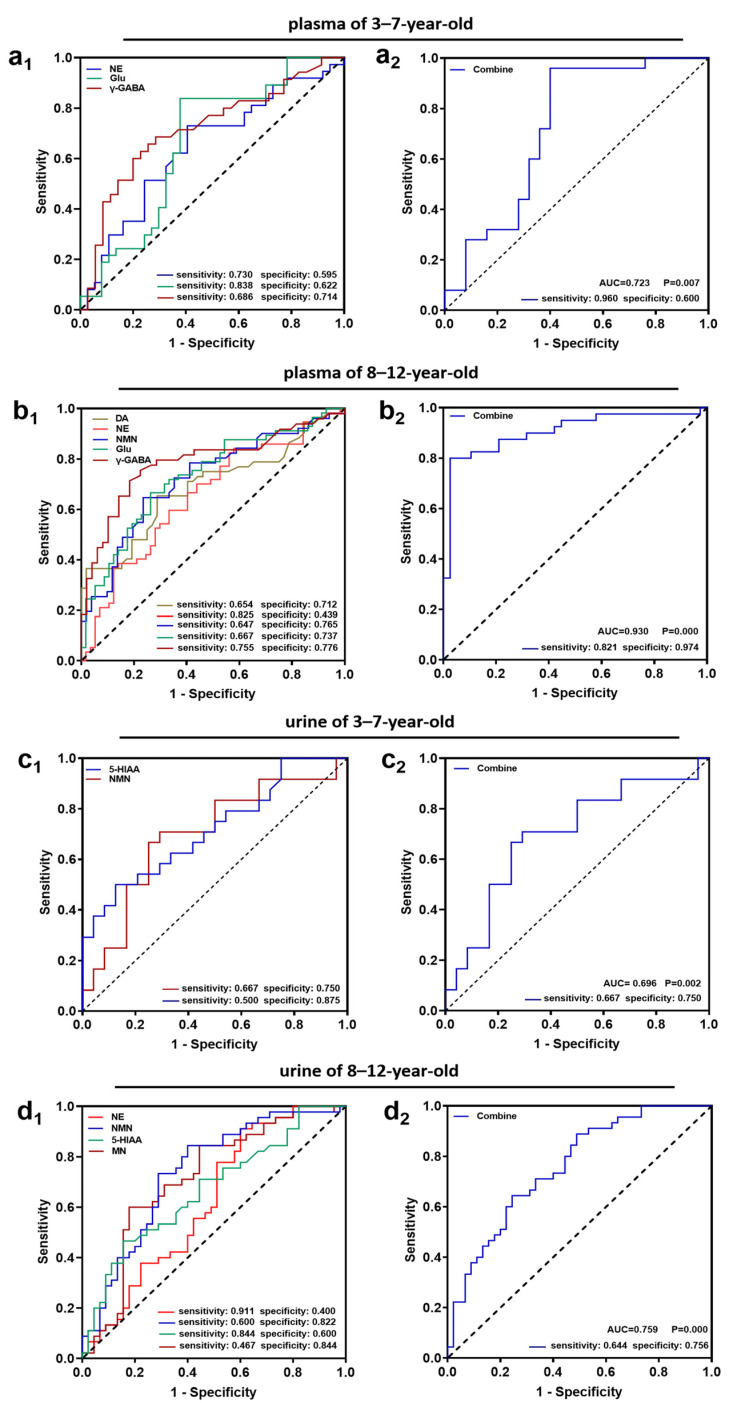
ROC curves of plasma and urine levels of neurotransmitters for predicting TD. (**a_1_**–**d_1_**) ROC curves of neurotransmitters with areas under the curve (AUCs) greater than 0.5 and significant differences (*p* < 0.05). (**a_2_**) ROC curve of the combination of NE, Glu, and γ-GABA levels in plasma from 3–7-year-old children. (**b_2_**) ROC curve of the combination of DA, NE, NMN, Glu, and γ-GABA levels in plasma from 8–12-year-old children. (**c_2_**) ROC curve of the combination of NMN and 5-HIAA levels in urine from 3–7-year-old children. (**d_2_**) ROC curve of the combination of NE, NMN, 5-HIAA, and MN levels in urine from 8–12-year-old children.

**Table 1 brainsci-12-00880-t001:** The number of plasma and urine samples in patients in the TD and control groups.

	Number of Plasma Samples		Number of Urine Samples	
	3–7-Year-Old	8–12-Year-Old	3–7-Year-Old	8–12-Year-Old
**TD**	37	57	24	45
**Control**	37	57	24	45

**Table 2 brainsci-12-00880-t002:** Neurotransmitter levels in plasma in patients in the TD and control groups.

Variable (nmol/L)	3–7-Year-Old			8–12-Year-Old		
	TD	Control	*p*	TD	Control	*p*
Dopamine (DA)	0.089 (0.056–0.225)	0.090 (0.029–1.417)	0.844	0.098 (0.061–0.250)	0.079 (0.038–0.127)	**0.001**
Epinephrine (E)	0.412 (0.107–2.106)	0.438 (0.114–1.323)	0.991	0.508 (0.105–1.762)	0.518 (0.139–1.032)	0.650
Norepinephrine (NE)	2.359 (0.515–6.164)	1.466 (0.536–4.824)	**0.035**	2.115 (0.593–9.338)	1.279 (0.527–8.947)	**0.005**
Normetanephrine (NMN)	0.203 (0.070–0.503)	0.151 (0.055–0.357)	0.184	0.249 (0.062–0.858)	0.138 (0.055–0.542)	**0.000**
5-hydroxytryptamine (5-HT)	323.000 (67.500–830.560)	320.000 (64.600–2564.840)	0.339	283.000 (26.400–1436.600)	262.000 (61.400–2041.600)	0.661
Glutamic Acid (Glu) (×10^4^)	5.483 (2.138–15.442)	3.603 (1.007–13.879)	**0.017**	6.053 (2.014–14.478)	3.541 (1.251–8.440)	**0.000**
Tyrosine (Tyr) (×10^4^)	6.938 (3.774–16.358)	6.515 (3.463–14.069)	0.440	6.840 (3.580–13.909)	7.813 (4.229–12.455)	0.080
Tryptophan (Trp) (×10^4^)	4.737 (2.830–7.570)	4.709 (2.304–8.087)	0.677	4.383 (2.099–7.417)	5.708 (3.080–11.129)	**0.000**
γ-aminobutyric acid (γ-GABA)	109.000 (35.400–840.100)	65.000 (16.850–442.226)	**0.002**	90.000 (21.600–785.000)	52.000 (13.400–145.800)	**0.000**

Values of patients with TDs and controls are shown as medians (2.5th percentile–97.5th percentile). A *p*-value < 0.05 indicates a significant difference.

**Table 3 brainsci-12-00880-t003:** Neurotransmitter levels in urine in patients in the TD and control groups.

Variable (µg/g crt)	3–7-Year-Old			8–12-Year-Old		
	TD	Control	*p*	TD	Control	*p*
Dopamine (DA)	465.095 (165.969–1131.521)	497.020 (174.262–1074.702)	0.967	388.200 (227.438–920.541)	403.610 (140.032–964.183)	0.768
Epinephrine (E)	35.860 (12.656–133.923)	60.375 (11.265–103.146)	0.918	20.580 (5.271–109.440)	48.930 (7.539–100.839)	**0.025**
Norepinephrine (NE)	47.180 (28.110–187.927)	42.320 (9.178–108.930)	0.322	44.260 (21.334–118.078)	37.470 (11.023–115.271)	**0.045**
Metanephrine (MN)	34.265 (11.174–159.292)	36.400 (5.761–135.284)	0.446	38.730 (12.427–106.105)	22.620 (7.600–112.756)	**0.000**
Normetanephrine (NMN)	200.635 (17.185–450.106)	91.355 (17.265–349.413)	**0.020**	92.820 (30.022–573.435)	51.560 (15.331–304.060)	**0.000**
Vanillylmandelic Acid (VMA) (×10^3^)	1.715 (0.418–8.883)	1.300 (0.279–5.483)	0.132	1.000 (0.314–13.543)	1.600 (0.187–9.141)	0.379
Homovanillic acid (HVA) (×10^3^)	5.010 (1.403–51.593)	4.226 (1.219–10.603)	0.279	2.540 (0.177–11.100)	3.470 (0.676–10.136)	**0.019**
5-Hydroxyindoleacetic acid (5-HIAA) (×10^3^)	5.810 (1.246–133.066)	2.220 (0.365–9.978)	**0.010**	2.960 (0.422–13.360)	1.750 (0.113–9.695)	**0.006**

Values of patients with TDs and controls are shown as medians (2.5th percentile–97.5th percentile). A *p*-value < 0.05 indicates a significant difference.

## Data Availability

The data presented in this study are available upon request from the corresponding author.

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
