# Peer review of "A Pilot Study on Plasma and Urine Neurotransmitter Levels in Children with Tic Disorders"

_brainsci, 2022, doi:10.3390/brainsci12070880_

Round 1

Reviewer 1 Report

Comments and Suggestions for Authors

well conducted study.

Author Response

Response:

Thanks for your comment.

Reviewer 2 Report

Comments and Suggestions for Authors

Thank you for providing me the opportunity to review this manuscript. The novelty behind this study is that it is complemented by the findings from the urine analysis. But overall , there are many previous reports about the profile of neurotransmitters in blood and/or CSF of TS individuals. Therefore, the authors should provide some evidence what new aspects, in comparison to previous reports, they present in their study. I have some further concerns about this manuscript. In particular, what is meant by this sentence: "The incidence has been increasing year by year" - do you mean that the incidence of TS is increasing with age? I do not agree since the usual onset is 6 years and the peak tic severity is around 12-13 or do you mean that there are some reports that TS is found more frequently than couple of years ago? I do not think it is true, maybe it is diagnosed more often? Could you please explain and provide citations if appropriate. Similarly, when talking about prevalence, could the authors provide citation of some other important studies, such as: https://pubmed.ncbi.nlm.nih.gov/25487709/, https://pubmed.ncbi.nlm.nih.gov/27548401/, https://pubmed.ncbi.nlm.nih.gov/22759682/ - these are all important studies that shoudl also be listed. It would also be appreciated to introduce some more information about previous studies about evaluation of urine and blood biomarkers in TS - this should be more ellaborated in the introduction. For the methods - could you please explain why some of the patients with tics were hospitalized? Did also the parent's give their informed consent? For the results - why in the first sentence the authors mention their unpublished study about sex differences? Similarly, why data are divided in two arbituary samples? Also, what TD is? Is this Tourette syndrome or tic disorders? All in all, the article needs significant revision - I can gladly reconsider it's publication after the appropriate changes are introduced. 

Author Response

Reviewer 2

Thank you for providing me the opportunity to review this manuscript. The novelty behind this study is that it is complemented by the findings from the urine analysis.

Response:

Thank you for your comments and your suggestions. All these comments and suggestions were taken into account as follows.

But overall , there are many previous reports about the profile of neurotransmitters in blood and/or CSF of TS individuals. Therefore, the authors should provide some evidence what new aspects, in comparison to previous reports, they present in their study.

Response:

Thanks for your comment and suggestions. Further literature investigation revealed that there were some previous reports about the profile of neurotransmitters in serum of TD or TS individuals. Therefore, we introduce and quote the relevant studies in the Introduction section, and explain the results in the relevant paragraphs in the Discussion section of our revised manuscript . (Line 55-57, 236-239, 250-252, 271-273, 291-294, Tracked change version)

We are very Sorry, we still did not find out studies about the profile of neurotransmitters in CSF of TD or TS individuals. Are the studies of neurotransmitter in CSF you see based on animal models? We speculate that this may be related to the fact that CSF tests require a lumbar puncture, a procedure that carries a risk of injury. And for ethical reasons, it is difficult to collect CSF for testing from patients with TD or healthy individuals. We have stated this view in the first paragraph of the Discussion section. (Line 205-212, Tracked change version)

I have some further concerns about this manuscript. In particular, what is meant by this sentence: "The incidence has been increasing year by year" - do you mean that the incidence of TS is increasing with age? I do not agree since the usual onset is 6 years and the peak tic severity is around 12-13 or do you mean that there are some reports that TS is found more frequently than couple of years ago? I do not think it is true, maybe it is diagnosed more often? Could you please explain and provide citations if appropriate. Similarly, when talking about prevalence, could the authors provide citation of some other important studies, such as: https://pubmed.ncbi.nlm.nih.gov/25487709/, https://pubmed.ncbi.nlm.nih.gov/27548401/, https://pubmed.ncbi.nlm.nih.gov/22759682/ - these are all important studies that shoudl also be listed.

Response:

Thank you for your comments and your suggestions. In the manuscript, the sentence: "The incidence has been increasing year by year" means that the occurrence of TD is more common in recent years than couple of years ago. This was based on the increased number of visits to our hospitals’ patients, and no relevant studies were reported to refer to. Therefore, it is unreasonable to say that the prevalence of TD increases year by year. Thank you for pointing out. We have modified the description of the TD prevalence based on your recommendations, and we have cited the very important studies that you have recommended in our revised manuscript. (Line 43-47, Tracked change version)

It would also be appreciated to introduce some more information about previous studies about evaluation of urine and blood biomarkers in TS - this should be more ellaborated in the introduction.

Response:

Thank you for your comments and your suggestions. We introduce and quote the relevant studies that about the profile of neurotransmitters in serum of TD or TS individuals in the Introduction section in our revised manuscript. (Line 55-57, Tracked change version)

For the methods - could you please explain why some of the patients with tics were hospitalized? Did also the parent's give their informed consent?

Response:

Thank you for your comments and your questions. In China, some patients who are refractory, or seriously affect the study and life, will be hospitalized. Our sample, with a small number, came from hospitalized patients. Therefore, in the patient information section, we described that the participants of patients attended our outpatient or inpatient visits. All subjects gave written informed consent prior to entering the study. Written informed consent was signed by their parents or guardians. (Line 79-96, Tracked change version)

For the results - why in the first sentence the authors mention their unpublished study about sex differences? Similarly, why data are divided in two arbituary samples?

Response:

Thank you for your questions. Our study team considered whether the level changes in neurotransmitters in patients with TD were related to sex and age. The study found that changes in neurotransmitter levels in TD patients were not related to sex, but to age. Therefore, we divided the participants aged 3-12-year-old into the 3-7-year-old and 8-12-year-old groups. According to the opinions of another reviewer, we adjusted the description of this part to the Materials and Methods section in our revised manuscript . (Line 97-106, Tracked change version)

Also, what TD is? Is this Tourette syndrome or tic disorders?

Response:

Thank you for your questions. TD means Tic disorders, TS means Tourette syndrome. The first sentence in the Introduction section gives a clear definition of these abbreviation, TD and TS. The second sentence has clarified the typing of patients with TD, including transient TD (TTD), chronic TD (CTD), and TS. (Line 41-44, Tracked change version)

All in all, the article needs significant revision - I can gladly reconsider it's publication after the appropriate changes are introduced.

Response:

Thanks for your comments and suggestions. All these comments and suggestions were taken into account. After the revision according to the comments, the manuscript has been significantly improved.

Reviewer 3 Report

Comments and Suggestions for Authors

Dear Authors,

First of all, I would like to congratulate for the theme of the manuscript, very original and interesting for the readers.

I would like to make some minor remarks that I have observed in the manuscript:

-The description of the participants should appear in the material and methods section. It would be advisable to include a small table with the descriptive data of the subjects in both the experimental and control groups.

-The sample procedures are described very extensively, so I would try to reduce them.

-Do the authors consider that systemic serotonin is the same as brain serotonin? 

Author Response

Reviewer 3

Dear Authors,

First of all, I would like to congratulate for the theme of the manuscript, very original and interesting for the readers.

Response:

Thanks for your comments.

I would like to make some minor remarks that I have observed in the manuscript:

-The description of the participants should appear in the material and methods section. It would be advisable to include a small table with the descriptive data of the subjects in both the experimental and control groups.

Response:

Thanks for your suggestions. We adjusted the description of the participants to the Materials and Methods section in our revised manuscript, and added a small table that included the descriptive data of the subjects in both the experimental and control groups. (Line 95-106, Tracked change version)

-The sample procedures are described very extensively, so I would try to reduce them.

Response:

Thanks for your suggestions. We have reduced the description of the sample procedures. We cite the relevant literature to help the readers further understand the details of sample procedures. (Line 118-122, Tracked change version)

-Do the authors consider that systemic serotonin is the same as brain serotonin?

Response:

Thank you for your question. We don’t think that brain serotonin is the same as systemic serotonin. However, the detection of brain serotonin requires the detection of levels in the CSF. CSF tests require a lumbar puncture, a procedure that carries a risk of injury. And for ethical reasons, it is difficult to collect CSF for testing from patients with TD or healthy individuals. We know that, neurotransmitters can move or penetrate the blood-brain barrier with the assistance of neurotransmitter transporters and are driven by electrochemical gradients. Therefore, blood neurotransmitter levels can indirectly reflect changes in neurotransmitter levels in the central nervous system (CNS). We have stated this view in the first paragraph of the Discussion section. (Line 205-212, Tracked change version)

Round 2

Reviewer 2 Report

Comments and Suggestions for Authors

Thank you for adresising my comments - I do not have further suggestions and recommend acceptance of this paper

Author Response

Thank you for your comment.